# A Novel EPANET Integration for the Diffusive–Dispersive Transport of Contaminants

Stefania Piazza, Mariacrocetta Sambito and Gabriele Freni *

School of Engineering and Architecture, University of Enna "Kore", Cittadella Universitaria, 94100 Enna, Italy
* Correspondence: gabriele.freni@unikore.it

**Abstract:** The EPANET model is commonly used to model hydraulic behaviour and water quality within water distribution networks. The standard version of the model solves the advective transport equation by solving a mass balance of the fundamental plug flow substance that considers the advective transport and kinetic reaction processes. Over the years, several versions of the model have been developed, which have made it possible to improve the modelling of water quality through the introduction of additional terms within the transport equation to solve the problem of dispersive transport (EPANET-AZRED) and to consider multiple interacting species in the mass flow and on the pipe walls (EPANET multi-species extension). The present study proposes a novel integration of the EPANET-DD (dynamic-dispersion) model, which enables the advective–diffusive–dispersive transport equation in dynamic flow conditions to be solved in the two-dimensional case, through the classical random walk method, implementing the diffusion and dispersion equations proposed by Romero-Gomez and Choi (2011). The model was applied to the University of Enna "KORE" laboratory network to verify its effectiveness in modelling diffusive–dispersive transport mechanisms in the presence of variable flow regimes. The results showed that the EPANET-DD model could better represent the actual data than previously developed versions of the EPANET model.

**Keywords:** EPANET; EPANET-DD; water distribution network; random walk method; water quality modelling



## 1. Introduction

EPANET is the model widely used to simulate the hydraulic behaviour and water quality within water distribution networks. This model was first developed by Rossman in 1994 [1,2] and distributed by the Environmental Protection Agency (EPA). This first version made it possible to solve the system of hydraulic equations implemented in the model (equation of continuity at nodes and pipes, equations of motion) through the method of the "gradient algorithm" [3]. As regards the solution of the water quality equation (advective-reactive transport equation), the conservation equation of the mass of the substance has been solved through the discrete volume element method (DVEM) [4], in which the mass substance is assigned to discrete volume elements once all connections in the network have been partitioned. This way, the concentration within each volume segment is first reacted and transferred to the adjacent downstream part. Suppose the latter is a junction node; the incoming mass and flow volumes mix with those already on the network nodes. Once these processes have been exhausted for all the network elements, the concentration is calculated and released in the first sections of the pipeline with flow out of the node.

In 2000, the model was updated to version 2.0 [5], in which the update concerns the water quality simulation section. Indeed, if in version 1.1, an Eulerian approach was used to solve the advective transport equation (DVEM), in this case, a time-based Lagrangian approach was used to track the fate of discrete water particles as they move along the tubes and mix at the junctions between time phases of fixed length. This significantly reduces the time passage of water quality compared to the hydraulic time passage, as the process is

linked to the speed of the particles and no longer to the flow rate passage. A further novelty of the EPANET2 model concerns the possibility of choosing between four mixing models inside the storage tank (complete mixing, two-compartment mixing, FIFO (first in, first out) plug flow, LIFO (last in, last out) plug flow), which in the first version was only plug flow.

Furthermore, in the same year, the EPA developed and distributed a 2.2 version of the model, in which only the hydraulic simulation section was updated, keeping the quality analysis section unchanged. This update concerned introducing two ways to model the water demand within the network junction nodes. In fact, until now, the water demand was modelled through demand driven analysis; that is, the water requests were fixed values that had to be supplied independently of the nodal pressures, and the connection flows produced by a hydraulic solution. This type of analysis can generate a paradox, as water demands may be met in nodes with negative pressures. To overcome this, the EPANET 2.2 model allows water demand modelling through pressure driven analysis [6]. The actual demand supplied to the node depends on the node's pressure. In fact, below a certain minimum pressure, the demand is zero; above a certain service pressure, the entire demand is supplied, and in between, the demand varies according to the power law of the pressure.

Regarding the water quality analysis, the above versions are limited exclusively to monitoring the advective transport and fate of a single chemical species, such as fluoride that can be used in a tracer study or free chlorine to study the decay of disinfectants. To consider multiple and interacting chemical species, in 2011, an extension of the EPANET model called multi-species extension (MSX) [7] was developed, capable of modelling the behaviour of any chemical species within water networks. This extension allows for the analysis of the redox phenomena that are generated between chlorine and natural organic matter (NOM), which constitutes a heterogeneous compound, but which in previous versions of the model was considered a constant, and the behaviour of some compounds such as chloramines which by their nature could not be modelled with simplified models.

To consider dispersive processes, which are intrinsic to transport mechanisms and relevant in the case of laminar flow regimes [8], researchers at the University of Arizona have developed an extension of the model called AZRED, introducing the processes of axial dispersion inside pipelines with low flow rates and considering the effects due to incomplete mixing for different types of junctions (cross, double-tee, and wye junction) [9–11].

The present study proposes a new version of the EPANET-DD (dynamic dispersion) model, in which diffusive–dispersive processes in two dimensions (axial and transverse) for laminar flow regimes have been considered. The model arises from a need highlighted in the study by Hart et al. (2016) [12], that is to study numerically under laminar flow conditions and with variable accelerations of the flow, how the particles of contaminant move inside the pipes, to determine the cause of the onset of double peaks. The authors argue that they could be caused by the different speeds running between the central part of the pipeline and at the edge near the wall. The EPANET-DD model has been implemented on MATLAB and solves the solute transport equation under dynamic conditions through the classical random walk method [13]. The latter was compared with the previous versions of the model (EPANET and AZRED) to evaluate its effectiveness in modelling the behaviour of the solute inserted within the laboratory network of the University of Enna "Kore".

## 2. Materials and Methods

### 2.1. Case Study

The Environmental Hydraulics Laboratory network of the University of Enna "KORE" (UKE) is a 1:1 scale distribution network, having a PN (nominal pressure) of 16 bar, DN (nominal diameter) of 63 mm, and a thickness of 5.8 mm (Figure 1).

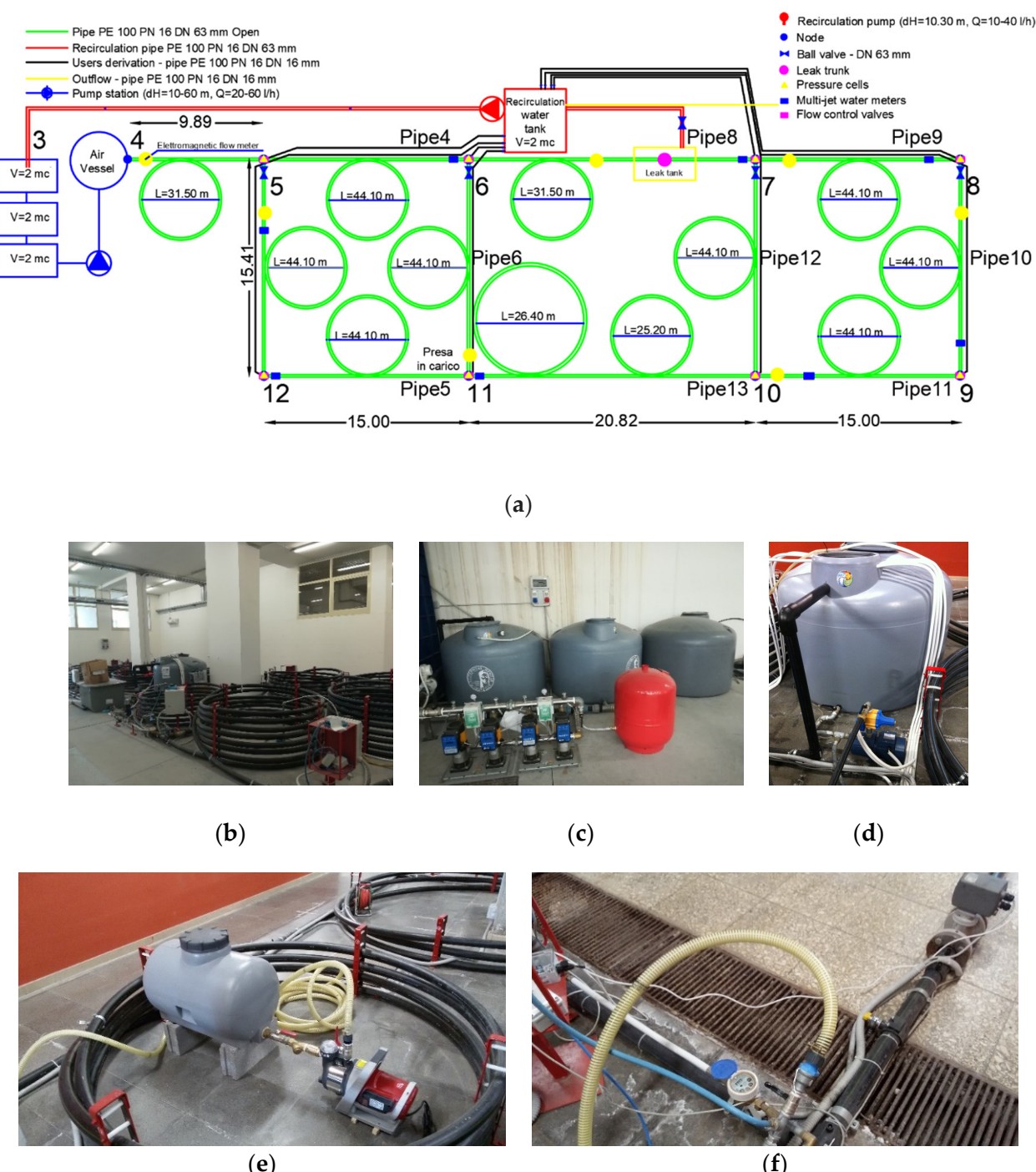

(**a**)

(**b**)                    (**c**)                    (**d**)

(**e**)                              (**f**)

**Figure 1.** The layout of the water distribution network (**a**), overview of the water distribution network (**b**), pumping system (**c**), recirculation system (**d**), installation of the pump-reservoir system (**e**) and connection to the node (**f**).

The network is divided into three meshes, each of which contains pipe windings with a radius of 2.0 m and a length of approximately 45 m. It is fed by three tanks with a capacity of 8 cubic meters employing a pumping system which allows the regulation of the pressure in the network between 1–6 bar. Furthermore, in the central part of the network, there is an additional tank in which the flows tapped at the user nodes are conveyed, and these are returned to the upstream tanks using a recirculation pump (Figure 1). The network has electromagnetic flow meters, pressure cells, multi-jet water meters, flow control valves and

a real-time water quality monitoring system (see [8] for further information). The network was contaminated with a sodium chloride concentration equal to 4600 mg/L injected into the upstream node (node 5), subject to an operating pressure of 1.5 bar, through a 100-litre tank and an injection pump. The simulations and experiments were carried out considering all open network branches.

### 2.2. EPANET-DD Model

The EPANET-DD model solves the equations under quasi-steady flow conditions, solving the hydraulic problem under steady flow conditions with the EPANET-MATLAB-Toolkit (Eliades, et al., 2016) [14] and the advection-diffusion-dispersion equation under dynamic flow conditions in the two-dimensional case with the classical random walk method [13], implementing the diffusion and dispersion equations proposed by Romero-Gomez and Choi (2011) [11].

The Toolkit uses an object-oriented approach, through the definition of a MATLAB class called epanet, which provides a standardised way to manage the network structure, to reach all functions as well as procedures that use multiple functions, to simulate and generally perform different types of network analysis, through the corresponding object. Inside there are local functions that make direct calls to EPANET. Through the function getComputedHydraulicTimeSeries, it was possible to perform the hydraulic simulation, solving the flow continuity and headloss equations recalled by EPANET.

Using the functions shown in Table 1, it was possible to obtain the speed, flow rate and headloss values on the pipes and the pressure, hydraulic head and actual demand values on the nodes.

**Table 1.** List of some Matlab Class Functions.

| Functions | Descriptions |
| --- | --- |
| getLinkVelocity | Current computed flow velocity (read only) |
| getLinkFlows | Current computed flow rate (read only) |
| getLinkHeadloss | Current computed head loss (read only) |
| getNodeHydaulicHead | Retrieves the computed values of all hydraulic heads |
| getNodeActualDemand | Retrieves the computed value of all actual demands |
| getNodePressure | Retrieves the computed values of all node pressures |

The flow continuity (Equation (1)) and headloss (Equation (2)) equations are solved using the "Gradient Method" developed by Todini and Pilati (1987) [2]. The first equation is solved around all nodes through Equation (1), in which $D_i$ is the flow demand at node I, and by convention, $Q_{ij}$ is the positive flow from a pipe $ij$ into the node.

$$\sum_j Q_{ij} - D_i = 0 \quad for\ i = 1, \ldots, N \tag{1}$$

The flow–headloss relationship in a pipe between nodes $i$ and $j$ are calculated as follow:

$$H_i - H_j = h_{ij} = rQ_{ij}^n - mQ_{ij}^2 \tag{2}$$

in which $H$ is the nodal head, $h$ is the headloss, $r$ the resistance coefficient, $Q$ the flow rate, $n$ is the flow exponent, and $m$ is the minor loss coefficient. The value of the resistance coefficient will depend on which friction headloss formula is used, defined with the formula of Hagen–Poiseuille, of Colebrook–White and cubic interpolation from the Moody diagram, as a function of the flow rate.

The quality analysis was developed using the classical random walk method. As demonstrated in the literature ([13–15]), the use of this combined method is possible due to the similarities between the Fokker–Planck–Kolmogorov equation and the advection-dispersion equation. The two equations are identical unless there is a conceptual difference between the parameters of the two equations, as the parameters present in the Fokker–

Planck–Kolmogorov equation are independent of time, resulting from the stationary hypothesis. To overcome this problem and address the issues related to discontinuities that could cause local mass conservation errors, [16], Delay et al. (2005) [13] provided a new equivalence, making this analogy valid again. This methodology can be easily applied to any flow model because the mass of the solute is discretised and transported by the particles in the random walk. Consequently, the mass conservation principle is automatically satisfied because the particles cannot suddenly disappear.

This model allows us to determine the position of the solute particles that move inside the network in the $x$ and $y$ directions as a function of the different flow regimes that occur inside the network, as shown in Equations (3) and (4):

$$x = x + \frac{3}{2} u_x \left( 1 - \left( \frac{y}{\frac{d}{2}} \right)^2 \right) dt + \sqrt{2 \cdot E_{f \text{ or } b} \cdot dt} \tag{3}$$

$$y = y + u_y dt + \sqrt{\left( E_f + E_b \right) \cdot dt} \tag{4}$$

where $ux$ corresponds to the component along the $x$ axis of the flow velocity, $uy$ equals to the component along the $y$ axis of the flow velocity, $dt$ is the duration of the contamination event, $d$ is the pipe diameter, and $Ef$ and $Eb$ are the forwards and backwards diffusion coefficients, respectively, as defined by Romero-Gomez and Choi (2011). In Equation (3), the diffusion coefficient assumes the forward or backwards values depending on whether the flow direction is positive or negative. The above equation was developed considering laminar flow conditions, in which the velocities in the network are relatively low. This allows the particles to move freely along the $y$ axis. This characteristic is also highlighted by the presence of the term in round brackets, $\left( 1 - \left( \frac{y}{\frac{d}{2}} \right)^2 \right)$, which multiplies the x component of the velocity $ux$. In fact, as the velocity along the $x$ direction increases and the flow rate changes, the particles tend to move along the preferred flow direction, and the term in brackets disappears from the equation.

To confine the particles inside the pipe section, the previous equations are solved considering the following boundary conditions (Equations (5) and (6)).

$$y = -2 \cdot y_{max} - y \quad for \quad y < -y_{max} \tag{5}$$

$$y = 2 \cdot y_{max} - y \quad for \quad y > y_{max} \tag{6}$$

where the particle position along $y$ is limited above and below by the physical presence of the pipe wall. The parameters $-ymax$ and $ymax$ coincide with the value of the pipe radius and take on a positive and negative value since the $x$ axis has been placed at the centre of gravity concerning the cross-section of the pipe. Using these two boundary conditions, the particles are not only prevented from escaping from the pipe but are also reflected, which prevents the particles from settling along the wall. These conditions are called the boundary reflection condition.

At this point, the contaminant concentration has been determined through Equation (7), in which the concentration value at the previous time has been increased by an amount that corresponds to the concentration per unit of particles $(C \cdot n)$ passing through the control volume $\left( \frac{L}{\Delta x} \cdot \pi \frac{d^2}{4} \right)$, where $L$ is the pipe length, $\Delta x$ is the section number of the pipe, and $\pi \frac{d^2}{4}$ is the cross-sectional area of the pipe.

$$C = C + \frac{C \cdot n}{\frac{L}{\Delta x} \cdot \pi \frac{d^2}{4}} \tag{7}$$

The three models (EPANET, AZRED and EPANET-DD) have been adequately calibrated both from a hydraulic and quality point of view.

The roughness coefficient was calibrated according to the flow rate measured upstream of the network (1.44 m³/h) and the diameter of each pipeline, calculating and iterating the uniform flow rate to coincide with the measured flow rate upstream of the network. Numerous experimental tests were conducted on the network, varying the pressure set at the pumping system (3.5–4.5 bar) and the flow rates drawn from the network nodes (between 5 and 15 L/min for nodes 5, 8 and 11).

Tables 2–4 show the calibrated roughness values of the pipes and the standard deviation (σ) values determined for the pressures at nodes 6, 7, 9, 10, the flow rates flowing into the network and the flow rates tapped at the nodes 5, 8, 11. Standard deviation of zero means that there is no variability between the data.

**Table 2.** Standard deviation between the pressures measured in the network and simulated numerically.

|  | Node 6 | Node 7 | Node 9 | Node 10 |
|---|---|---|---|---|
| σ [mH$_2$O] | 0.01 | 0.15 | 0.05 | 0.09 |

**Table 3.** Pipes roughness and standard deviation between the flow rates measured in the network and simulated numerically.

|  | Link 5 | Link 6 | Link 7 | Link 9 | Link 10 | Link 11 | Link 13 |
|---|---|---|---|---|---|---|---|
| Roughness [mm] | 1 | 1 | 1 | 1 | 1 | 1 | 1 |
| σ [m³/h] | 0.12 | 0.12 | 0.08 | 0.11 | 0.11 | 0.11 | 0.15 |

**Table 4.** Standard deviation between the measured tapped flow rates and the numerically simulated flow rates.

|  | Node 5 | Node 8 | Node 11 |
|---|---|---|---|
| σ [L/min] | 0.45 | 0.07 | 0.07 |

The backward and forward dispersion coefficients ($E_b = 0.17$ and $E_f = 0.51$, respectively) were calibrated through a trial-and-error operation using statistical parameters such as Nash–Sutcliffe efficiency (*NSE*) [17], Kling–Gupta efficiency (*KGE*) [18] and coefficient of determination (R$^2$) [19].

The Nash–Sutcliffe efficiency (*NSE*) coefficient [17] is a hydrology metric that measures how well a model simulation predicts an outcome variable. It is defined as one minus the ratio of the error variance of the modelled time series divided by the variance of the observed time series, as shown in Equation (8):

$$NSE = 1 - \frac{\sum(y_i - y_{i,\,sim})^2}{\sum(y_i - \overline{y})^2}$$
(8)

where $y_i$ and $y_{i,\,sim}$ correspond to the measured and simulated values of the variable, respectively, and $\overline{y}$ is the average of the measured values of $y$. If *NSE* = 1, there is a perfect correspondence between the model and the observed data; if *NSE* = 0, the model has the same predictive capacity as the average of the time series in terms of the sum of the square errors. If *NSE* < 0, the observed mean is a better predictor of the model.

The Kling–Gupta efficiency (*KGE*) coefficient [18] is a metric that measures the goodness of fit (Equation (9)). It consists of three main components: the correlation coefficient between the observations and simulations $r$, the ratio between the standard deviation of

the simulated values and the standard deviation of the observed values, and the balance between the average of the simulated values and the average of the experimental values.

$$KGE = 1 - \sqrt{(r-1)^2 + \left(\frac{\sigma_{sim}}{\sigma_{obs}} - 1\right)^2 + \left(\frac{\mu_{sim}}{\mu_{obs}} - 1\right)^2} \tag{9}$$

Similar to the *NSE* coefficient, *KGE* = 1 indicates perfect agreement between the simulations and observations. For *KGE* values <= 0, analogous to what the authors observed for *NSE* values, all negative values below the threshold *KGE* = 0 indicate results with poor model performance.

The coefficient of determination ($R^2$) measures the goodness of fit of a statistical model. It is defined as the squared value of the linear correlation coefficient. The $R^2$ value ranges between 0 and 1. A value of zero indicates that there is no correlation between the two data series. On the other hand, higher coefficient values indicate a better fit for the model. However, it is not always true that large $R^2$ values result in a good model fit, as the linear correlation coefficient could produce a perfectly positive or negative relationship [19].

### 3. Results and Discussion

Figure 2 compares the experimental results (dotted line), obtained by considering all the branches of the network open and the three different modelling approaches (advective model: grey line; Romero-Gomez and Choi model: dashed line; and EPANET-DD model: continuous line), obtained by contaminating the UKE network at node 5 with a sodium chloride concentration of 4600 mg/L for 3 min, monitoring nodes 6 (a), 7 (b), 8 (c), 9 (d), 10 (e) and 11 (f) of the UKE network and evaluating the effect generated by the different Reynolds values present in the network. The test was 3 h, and the flow rates tapped at the nodes were set to achieve different flow regimes in the network, as shown in Table 5. It was observed that the three flow regimes (laminar, transition and turbulent) occurred simultaneously within the network as a function of the tapped flow rates. In particular, there was a turbulent flow regime at node 6, which is immediately downstream of the contaminant inlet node. At greater distances from the contaminant entry node, the authors observed a variation in the flow regime, oscillating between transition and laminar flow. Furthermore, some network nodes were affected by both flow regimes (Figure 2e,f) as they converged to two network branches (Figure 1).

The numerical analysis was carried out considering the following calibration coefficient values: backwards and forwards coefficients equal to 0.17 and 0.51, respectively. These coefficients provided a better fit between the simulated and measured data.

It was observed that the advective model could not represent all the experimentally monitored nodes. This demonstrates the inadequacy of the advective model in reproducing experimental data. Although some nodes had high values for the parameters *KGE*, *NSE* and $R^2$, graphically, there is no correspondence between the simulated and measured data to justify these values. As shown in Figure 2, this model worked well only for one node of the network (Figure 2a), which is supported by a high value of the *KGE*, *NSE* and $R^2$ coefficients reported in Table 6. This node, directly downstream of the node where sodium chloride was injected, was characterised by a turbulent flow regime with a Reynolds number of 4112. Under these conditions, the advective model could centre the contamination's peak and the time interval in which it occurred, despite having a higher peak concentration. Analysing the remaining nodes shown in Figure 2b–f, the advective model reproduced anticipated events concerning the experimental data, which in some cases corresponded to a few minutes. Still, in the case of Figure 2c, the event was expected for approximately an hour. Furthermore, as shown in Figure 2c,d, the model underestimated the persistence of the contamination, as the event was quickly exhausted.

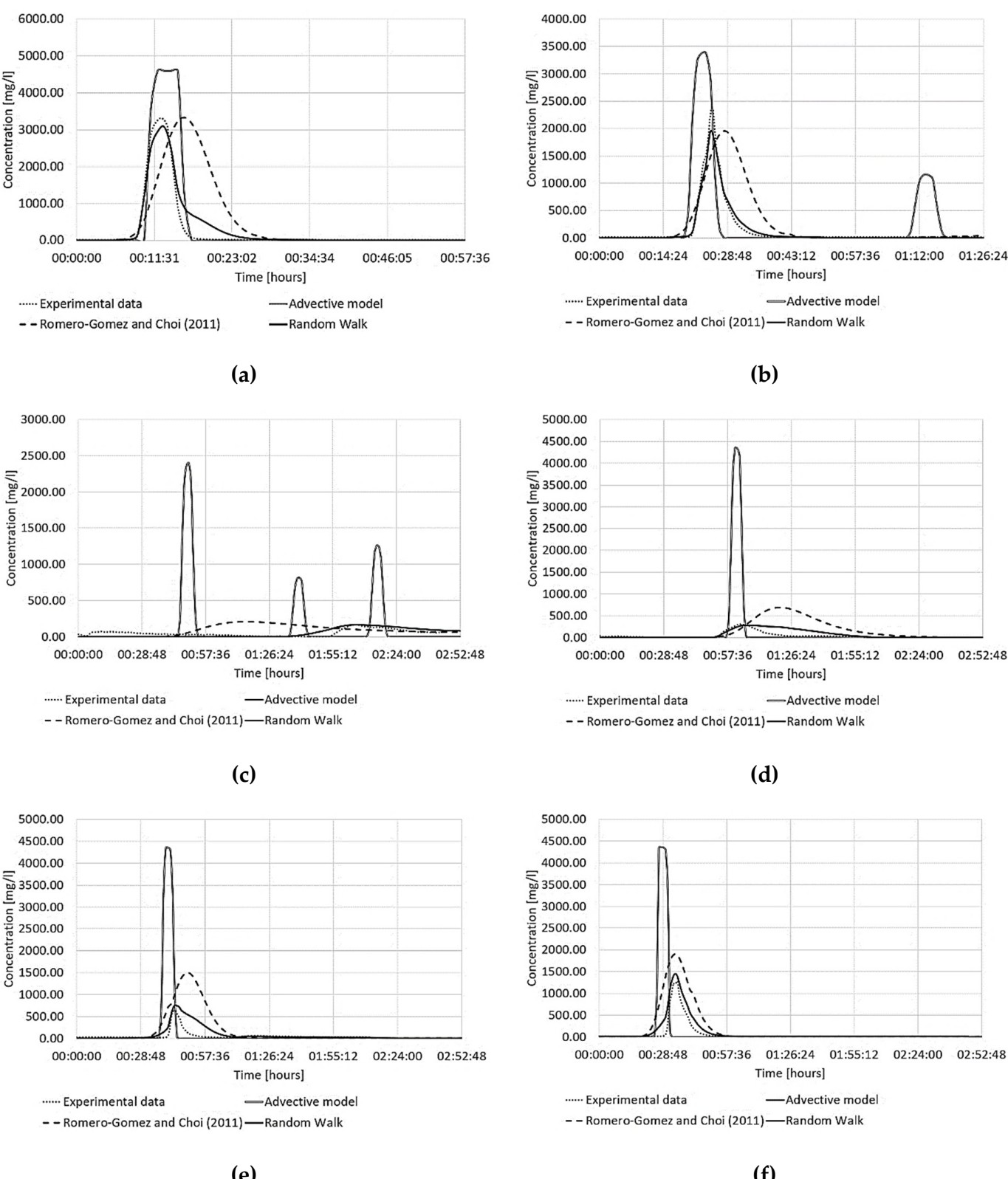

**Figure 2.** Comparison between experimental and simulated data using the three models (advective, Romero-Gomez and Choi (2011) [11] (backwards coefficient = 0.17, forwards coefficient = 0.51), EPANET-DD (backwards coefficient = 0.17, forwards coefficient = 0.51)) for a contamination event lasting 3 h subject to a network pressure of 1.5 bar for nodes 6 (**a**), 7 (**b**), 8 (**c**), 9 (**d**), 10 (**e**), 11 (**f**), obtained by contaminating node 5.

The results of the advective-dispersive model, which is based on the formulations of Romero-Gomez and Choi, obtained using the previously calibrated backwards and forwards coefficients differed more from other models and the trend of the experimental data. As shown in Figure 2, although the model could reproduce the dispersive effect of the contaminant in the network, it could not reproduce the behaviour of the experimental data. This is easily seen for the nodes in Figure 2c–e, where the model overestimated the contamination peaks and the persistence of the contaminant within the distribution network and reproduced a significantly anticipated event in Figure 2c but delayed events for all other nodes. The model performed slightly better for the node in Figure 2f, which had a higher coefficient of determination $R^2$ than that calculated for the other nodes. In this node, the coexistence of the transition and laminar flow regimes, which had Reynolds values of 3598 and 200, respectively, was observed. The transition flow regime most likely dominated because the dispersive effects were not overpowering. Furthermore, the model could centre the contamination peak for the node above despite having overestimated its mass.

Using the same backwards and forwards coefficients used in the Romero-Gomez and Choi model, it was observed that the results obtained by applying the new EPANET-DD model outperformed the previous two models used in this study. This is supported by the high *KGE*, *NSE* and $R^2$ coefficient values, as shown in Table 6. In fact, for all the monitored nodes in Figure 2, the model could centre the contamination peak. Furthermore, considering Figure 2b,f, which have transition flow regimes with Reynolds numbers of 3598, the model perfectly fits the experimental data departing from the classic bell-shaped trend typical of a Gaussian distribution. It is worth noting that at node 9 in Figure 2d, which has a laminar flow regime with a Reynolds number of 514, the model perfectly fits the ascending branch of the experimental data but failed to reproduce the descending limb of the curve. This is due to a separation between the contamination behaviour at the edge and the centre of the pipeline caused by the transition from turbulent to laminar flow. For the previous node, in Figure 2e, the model had a gap with the experimental data in the descending part of the curve but perfectly reproduced the terminal part of the pollutograph.

Using a model capable of reproducing experimental data in the presence of laminar flow regimes and that is able to adequately reproduce the diffusive–dispersive transport mechanisms, enables the design of an effective and efficient early warning system, in order to safeguard consumer health. Furthermore, it allows the design of water networks to be improved, in order to avoid the re-tin phenomena not only at the dead-ends, in which the presence of a laminar flow regime is expected, but also at the neutral points within the meshed networks.

**Table 5.** Reynolds number and flow regime.

|  | Link 4 | Link 6 | Link 7 | Link 9 | Link 10 | Link 11 | Link 12 | Link 13 |
|---|---|---|---|---|---|---|---|---|
| Reynolds (Re) | 4112 | 200 | 3598 | 1542 | 514 | 2056 | 1542 | 3598 |
| Flow regime | Turbulent | Laminar | Transition | Laminar | Laminar | Transition | Laminar | Transition |

**Table 6.** Comparison of statistical parameters (Kling–Gupta efficiency, Nash–Sutcliffe efficiency, $R^2$) for the advective, Romero-Gomez and Choi (2011) and EPANET-DD models.

| Node | Advective Model | | | Romero-Gomez and Choi (2011) Model | | | EPANET-DD Model | | |
|---|---|---|---|---|---|---|---|---|---|
| | *KGE* | *NSE* | $R^2$ | *KGE* | *NSE* | $R^2$ | *KGE* | *NSE* | $R^2$ |
| 6 | 0.44 | 0.52 | 0.29 | −0.60 | −0.72 | 0.21 | 0.63 | 0.69 | 0.49 |
| 7 | 0.25 | 0.59 | 0.68 | −0.08 | −0.15 | 0.12 | 0.81 | 0.84 | 0.76 |
| 8 | −0.55 | −1.50 | 0.08 | 0.01 | 0.35 | 0.04 | 0.45 | 0.43 | 0.92 |
| 9 | 0.22 | 0.18 | 0.43 | −1.58 | −5.57 | 0.13 | 0.29 | 0.35 | 0.17 |
| 10 | 0.34 | −0.01 | 0.19 | −4.35 | −14.81 | 0.09 | −0.15 | −0.54 | 0.55 |
| 11 | −0.30 | −0.62 | 0.05 | −0.94 | −1.18 | 0.79 | 0.42 | 0.76 | 0.90 |

## 4. Conclusions

In this study, the new model called EPANET-DD (dynamic dispersion) was presented and performed on the University of Enna "Kore" laboratory network. The model performance was evaluated using three different coefficients (*KGE*, *NSE*, $R^2$). Two other models (the advective model EPANET and an advective-dispersive model based on the formulations of Romero-Gomez and Choi) were evaluated concerning the experimental data, and the performance of the new model was compared with the results obtained from the advective model.

In summary, the main conclusions of this study are as follows:

- The advective model works well only in locations close to the contamination node, where it can intercept the contamination's peak even for lower values. In fact, relatively high values of the *KGE*, *NSE* and $R^2$ coefficients were observed at node 6 near the contamination node (0.44, 0.52, 0.29 respectively).
- In all other cases, the contamination event was anticipated and had a shorter duration than that detected by the experimental campaign. As a result, much lower or even negative values of the three coefficients were obtained.
- The Romero-Gomez and Choi model can represent the dispersive behaviour of the contaminant. Still, it poorly represents the experimental data regarding delay or anticipation of the contamination peak and overestimating the contaminant mass. This was confirmed by the coefficients *KGE*, *NSE*, $R^2$ which resulted in some nodes (6, 7, 9, 10) being worse than those obtained using the advective model.
- The new EPANET-DD model produced the best results in terms of adaptability with the experimental data. It simultaneously represented the peak time and provided better accuracy than the Romero-Gomez and Choi model. In fact, the coefficients considered were very high and, in some cases, close to unity.

Furthermore, taking into account the laminar flow regime and the diffusion–dispersion processes, it is possible to better position the early warning sensors and better design the networks to avoid stagnation phenomena not only in the dead ends where the laminar flow regime is expected but even in the neutral points of the closed mesh.

**Author Contributions:** G.F. designed the research; G.F. wrote the code; S.P., M.S. and G.F. performed the analysis; S.P. analysed the data; and S.P., M.S. and G.F. wrote the paper. All authors have read and agreed to the published version of the manuscript.

**Funding:** This research received no external funding.

**Institutional Review Board Statement:** Not applicable.

**Informed Consent Statement:** Not applicable.

**Data Availability Statement:** https://github.com/gabrielefreni/epanet_dd.git, accessed on 23 August 2022.

**Conflicts of Interest:** The authors declare no conflict of interest.

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
