# Peer review of "A Novel EPANET Integration for the Diffusive–Dispersive Transport of Contaminants"

_water, doi:10.3390/w14172707_

Round 1

Reviewer 1 Report

Authors propose a novel integration of the EPANET-DD (Dynamic-Dispersion) model to solve the advective- diffusive-dispersive transport equation in dynamic flow conditions in the two-dimensional case, using the classical random walk method.

The work is well organized and very clearly presented.

Authors devoted much of the work to the introduction section. This way they well presented the evolution of the software they use, EPANET, making clear the reasons for the integration they propose.

The WDN used is well described and the layout support images are very helpful for understanding.

The reviewer asks to check some points in the text:

1) In line 110 Figure 1 is repeated, I believe it should be eliminated

2) Line 153 presents a variation of the font

3) I recommend also inserting the points where the devices are positioned in the network layout (devices listed in line 110-115)

4) Do the authors believe that the backward and forward dispersion, calibrated through a trial-and-error operation, can easily be carried out for medium-sized real networks? (line 206)

5) Part of Figure 1 is shown in line 250. The caption reads Figure 1, but I believe this is a mistake.

Author Response

The authors propose a novel integration of the EPANET-DD (Dynamic-Dispersion) model to solve the advective- diffusive-dispersive transport equation in dynamic flow conditions in the two-dimensional case, using the classical random walk method.

The work is well organized and very clearly presented.

Response: Thanks to the reviewer for the comments.

Authors devoted much of the work to the introduction section. This way they well presented the evolution of the software they use, EPANET, making clear the reasons for the integration they propose.

Response: Thanks to the reviewer for the comments.

The WDN used is well described and the layout support images are very helpful for understanding.

Response: Thanks to the reviewer for the comments.

The reviewer asks to check some points in the text:

1) In line 110 Figure 1 is repeated, I believe it should be eliminated

Response: The repetition was eliminated.

2) Line 153 presents a variation of the font

Response: typographic issues were solved

3) I recommend also inserting the points where the devices are positioned in the network layout (devices listed in line 110-115)

Response: Devices have been added to the layout.

4) Do the authors believe that the backward and forward dispersion, calibrated through a trial-and-error operation, can easily be carried out for medium-sized real networks? (line 206)

Response: Thanks for the question. Yes, the Romero-Gomez-Choi approach (only axial dispersion but including backward and forward coefficients) was applied, in a previous study (doi:10.2166/ws.2019.131), to the real network of Zandvoort in Netherlands using the same trial-and-error operation for the calibration and satisfactory results were obtained.

5) Part of Figure 1 is shown in line 250. The caption reads Figure 1, but I believe this is a mistake.

Response: The error has been corrected.

Reviewer 2 Report

The present study proposes an integration of the EPANET-DD (Dynamic-Dispersion) model, which allows solving the advective-diffusive-dispersive transport equation in dynamic flow conditions in the two-dimensional case, through the classical random walk method. The model was applied to the University of Enna "KORE" laboratory network to verify its effectiveness in modeling diffusive-dispersive transport mechanisms in the presence of variable flow regimes. The results showed that the EPANET-DD model could better represent actual data than previously developed versions of the EPANET model. The following remarks are to be considered: Line 110 and line 250: Figure 2 is a duplication of Figure 1. The layout of the water distribution network (a), overview of the water distribution network (b), pumping system (c), recirculation system (d), installation of the pump-reservoir system (e), and connection to the node (f). Add a reference to the equation, in case you are not the Author of it. What lessons should water companies draw from this analysis? Are there concrete steps that can be recommended and how generalizable are the findings? Some information about the practical use of the obtained results, both in the section Results and Conclusions should be underlined. What is the added value of the paper? Why the paper should be recommended? Conclusion is very general, it should be supported by the results. If possible, add the DOI numbers to the publications. 

Author Response

The present study proposes an integration of the EPANET-DD (Dynamic-Dispersion) model, which allows solving the advective-diffusive-dispersive transport equation in dynamic flow conditions in the two-dimensional case, through the classical random walk method. The model was applied to the University of Enna "KORE" laboratory network to verify its effectiveness in modeling diffusive-dispersive transport mechanisms in the presence of variable flow regimes. The results showed that the EPANET-DD model could better represent actual data than previously developed versions of the EPANET model. The following remarks are to be considered:

Line 110 and line 250: Figure 2 is a duplication of Figure 1. The layout of the water distribution network (a), overview of the water distribution network (b), pumping system (c), recirculation system (d), installation of the pump-reservoir system (e), and connection to the node (f).

Response: The repetition was removed.

Add a reference to the equation, in case you are not the Author of it. 

Response: References relating to the literature equations have been included in the text.

What lessons should water companies draw from this analysis?

Response: This study could help water companies to adequately simulate the transport mechanisms of contaminants even in the case of laminar flow regimes, in which the diffusive-dispersive component takes on considerable importance, to prevent possible contamination events, which are harmful to consumer's health.

Are there concrete steps that can be recommended and how generalizable are the findings?

Response: The model is applicable everywhere in the hypothesis of dissolved contaminants and the transition between laminar and turbulent flow regime is defined; for water managers, the model allows the areas affected by contamination to be defined with greater precision and makes emergency plans more effective if it is necessary to intervene on contamination that is hazardous to health. It allows you to design more effective strategies for cleaning a contaminated network. allows you to better manage the control in real-time to manage the flow variations in the network and always guarantee a turbulent flow and greater water exchange

Some information about the practical use of the obtained results, both in the section Results and Conclusions should be underlined.

Response: More information has been added.

What is the added value of the paper?

Response: Unlike the AZRED model, which considers only the axial dispersion, in this study the transverse component was also considered, which contributed significantly to improving the performance of the model.

Why the paper should be recommended?

Response: This model could be coupled with optimization methods for optimal sensor placement to prevent accidental or deliberate contamination events or to identify contamination sources, obtaining reliable results as the starting model can reproduce experimental data effectively.

Conclusion is very general, it should be supported by the results. 

Response: The conclusions have been improved.

If possible, add the DOI numbers to the publications. 

Response: Where possible, DOI numbers have been added.

Round 2

Reviewer 2 Report

the manuscript can be accepted